# Interpretable Semantic Role Relation Table for Supporting Facts Recognition of Reading Comprehension

## Abstract

The current Machine Reading Comprehension (MRC) model has poor interpretability. Interpretable semantic features can enhance the interpretability of the model. Semantic role labeling (SRL) captures predicate-argument relations, such as "who did what to whom," which are critical to comprehension and interpretation. To enhance the interpretability of the model, we propose the semantic role relation table, which represents the semantic relation of the sentence itself and the semantic relations among sentences. We use the name of entities to integrate into the semantic role relation table to establish the semantic relation between sentences. This paper makes the first attempt to utilize contextual semantic's explicit relation to the recognition supporting fact of reading comprehension. We have established nine semantic relation tables between target sentence, question, and article. Then we take each semantic relationship table's overall semantic role relevance and each semantic role relevance as important judgment information. Detailed experiments on HotpotQA, a challenging multi-hop MRC data set, our method achieves better performance. With few training data sets, the model performance is still stable.

## 1 Introduction

There has been an increasing interest in the explainability of Machine Reading Comprehension (MRC) in recent years. For enhancing the explainability in MRC, some researchers (Qiu et al., 2019; Tu et al., 2020; Fang et al., 2020) utilize Graph Networks. The relational inductive bias encoded in Graph Networks (Battaglia et al., 2018) provides viable support for reasoning and learning over structured representations. Some researchers (Feng et al., 2020) utilize explicit inference chains for multi-hop reasoning. Yang et al. (2018) provide sentence-level supporting facts required for reasoning, allowing QA systems to reason with strong supervision and explain the predictions. This paper focuses on establishing an interpretable model sentence-level supporting facts recognition. A system capable of delivering explanations is generally more interpretable, meeting some of the requirements for real-world applications, such as user trust, acceptance, and confidence (Thayaparan et al., 2020).

Figure 1: An example of the multi-hop questions in HOTPOTQA.The the supporting facts in blue.The red area is an entity that is highly relevant to the question.

An example from HotpotQA (Yang et al., 2018) is illustrated in Figure1. To correctly answer the question ("The director of the romantic comedy Big Stone Gap is based in what New York City") the model is required to first identify ParagraphB as a relevant paragraph, whose title contains the keywords that appear in the question ("Big Stone Gap"). S3, the first sentence of ParagraphB, is then chosen by the model as a supporting fact that leads to the next-hop paragraph ParagraphA. Lastly, from ParagraphA, the span Greenwich Village, New York City is selected as the predicted answer. In this example, s1 and s3 contain the critical information needed to reason the answer. When we judge whether S3 is a supporting fact, our model needs to understand the semantic relation between the S3,question and the paragraphs.

| S1 / S2 | who | eat | bread | at night |
|---|---|---|---|---|
| Tom | | | | |
| eat | | | | |
| bread | | | | |
| in the | | | | |
| morning | | | | |

| S1 / S2 | ARG0 | V | ARG1 | TMP |
|---|---|---|---|---|
| ARG0 | | | | |
| V | | | | |
| ARG1 | | | | |
| TMP | | | | |

Figure 2: a example of attention mechanism    Figure 3: a example of semantic relational table

People can identify the supporting factors in the paragraph and give a detailed explanation of the judgment result. We argue that the more specific the semantic interpretation, the more helpful the model imitates the human reasoning process. The input of the most recent model is Pre-trained embeddings, which have the advantage of capturing semantic similarity, but it is hard to explain in detail. We believe that the model uses interpretable features for reasoning, which contributes to enhanced interpretability. Recently, the attention mechanism has achieved remarkable performance on many natural language processing tasks. The model of attention mechanism learns the relevance between words through training(Figure 2). Inspired by the attention mechanism, for establishing rich and interpretable semantic features, we propose the semantic relational table(Figure 3).

Semantic role labeling (SRL) is a shallow semantic parsing task aiming to discover who did what to whom, when and why(He et al., 2018; Li et al., 2018), providing explicit contextual semantics, which naturally matches the task target of text comprehension and It is easy to explain for people.

Recently, Ribeiro et al. (2020) show that although measuring held-out accuracy has been the primary approach to evaluate generalization, it often overestimates the performance of NLP models. Moreover, the model does not seem to resolve basic Coreferences and grasp simple subject/object or active/passive distinctions(SRL), all of which are critical to comprehension. Zhang et al. (2018) regard the semantic signals as SRL embeddings and employ a lookup table to map each label to vectors, similar to the implementation of word embedding. For each word, a joint embedding is obtained by the concatenation of word embedding and SRL embedding. Extensive experiments on benchmark machine reading comprehension and inference datasets verify that the proposed semantic learning helps for the model. The previous work indicates that SRL may hopefully help to understand contextual semantics relation.

Formal semantics generally presents the semantic relation as "predicate argument" structure. For example, given the following sentence with target verb (predicate) sold, all the arguments are labeled as follows, [ARG0: Tom] [V: sold] [ARG1: a book] [ARG2: to jerry] [ARGM-TMP: last week]. Where ARG0 represents the seller (agent), ARG1 represents the thing sold (theme), ARG2 represents the buyer (recipient), ARGM-TMP is an adjunct indicating the timing of the action, and V represents the predicate.

A question sentence example: [ARG0: Who] [V: bought] [ARG1: flowers] [ARG2: for jerry] [ARGM-LOC:in the park] [ARGM-TMP: last week]. In the reference text, the sentences highly related to the semantic roles of the question are essential reasoning information. In this paper, we use the name of entities to integrate into the semantic role relation table to establish the semantic relation between sentences. Since many features are difficult to process efficiently, we simplified the semantic role relation table between two sentences.

In particular, the contributions of this work are: (1)We believe that the model uses interpretable features for reasoning, which contributes to enhanced interpretability. So we propose an interpretable form of semantic relations to enhance the interpretability of the model's input data. We use the enti-

ty's name to integrate into the semantic role relation table to establish the semantic relation between question, article, and the target sentence. (2)With few training data sets, the model's performance based on the semantic role relation table is still stable.

## 2 METHOD

### 2.1 A SIMPLIFIED EXAMPLE

For each sentence, we extract predicate-argument tuples via SRL toolkits. We employ the BERT-based model (Shi & Lin, 2019) in the AllenNLP toolkit to perform SRL. We arrange the different semantic roles labels in a fixed order. A simplified example of supporting sentence prediction task:

- question: who eats bread at night?
- sent1: Tom eats bread.
- sent2: Jerry eats an apple.
- sent3: Tom eats something at night.
- sent4: Jerry drinks milk.

We need to recognize sent1 and sent3 as the supporting facts. First step: we use semantic role label tools to parse sentences, set the position that the semantic role label contained in the sentence to 1, and set the position that the semantic role label is not contained in the sentence to 0. Second step: We match the entities between the two sentences, the position of the semantic role tag corresponding to the same named entities set to 1, and the matching result is regarded as the semantic relation between the sentences. The two steps can build the following semantic role relation tables: The "q" stands for "question",the "a" stands for "article",the "t" stands for "table".

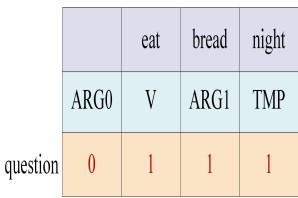
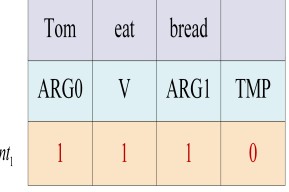
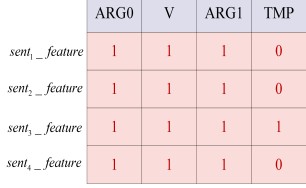

Figure 4: q_feature_t          Figure 5: sent1_feature_t          Figure 6: a_feature_t

The question_feature_table (Figure 4) shows the distribution of the semantic structure information of the question sentence. "Who" does not appear in the figure due to it is regarded as a stop word. The semantic role label feature of the question sentence is an essential clue for judging whether the sentence is a piece of evidence.

The sent1 is the target sentence for which the model makes evidence prediction. sent1_feature_table (Figure 5), show the distribution of the semantic structure information of the sentence. The article_feature_table(Figure 6) show the distribution of the global semantic structure information of the article.

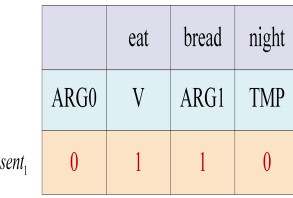
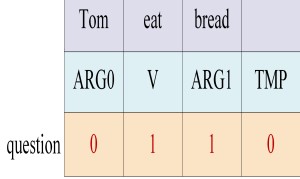
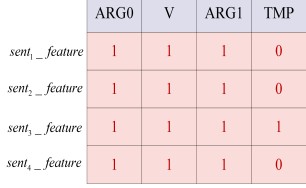

Figure 7: q_feature_sent1_t          Figure 8: sent1_feature_q_t          Figure 9: sent1_feature_a_t

The question_feature_sent1_table (Figure 7) and The sent1_feature_question_table (Figure 8) denotes the semantic relationship between the problem and the target sentence.

| | ARG0 | V | ARG1 | TMP |
|---|---|---|---|---|
| $sent_1\_feature$ | 1 | 1 | 1 | 0 |
| $sent_2\_feature$ | 0 | 1 | 0 | 0 |
| $sent_3\_feature$ | 1 | 1 | 0 | 0 |
| $sent_4\_feature$ | 0 | 0 | 0 | 0 |

Figure 10: a_feature_sent1_t

| | eat | bread | night |
|---|---|---|---|
| | ARG0 | V | ARG1 | TMP |
| $sent_1$ | 0 | 1 | 1 | 0 |
| $sent_2$ | 0 | 1 | 0 | 0 |
| $sent_3$ | 0 | 1 | 0 | 1 |
| $sent_4$ | 0 | 0 | 0 | 0 |

Figure 11: q_feature_a_t

| | ARG0 | V | ARG1 | TMP |
|---|---|---|---|---|
| $sent_1\_feature$ | 0 | 1 | 1 | 0 |
| $sent_2\_feature$ | 0 | 1 | 0 | 0 |
| $sent_3\_feature$ | 0 | 1 | 0 | 1 |
| $sent_4\_feature$ | 0 | 0 | 0 | 0 |

Figure 12: a_feature_q_t

The sent1_feature_article_table(Figure 9) and article_feature_sent1_table(Figure 10) denotes the semantic relationship between the article and the target sentence.

The question_feature_article_table(Figure 11) and article_feature_question_table(Figure 12) denotes the semantic relationship between the article and the question.

## 2.2 MODEL

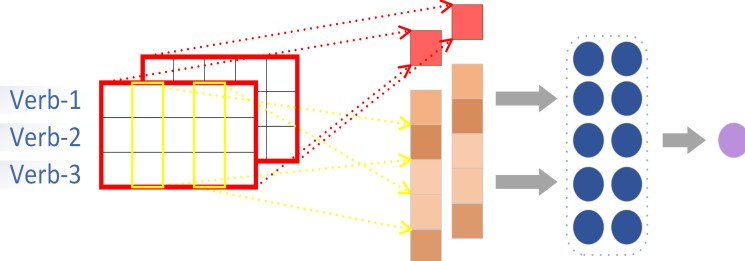

Figure 13: There are 9 semantic role relation tables. In this figure, we use two tables of a sentence size. Each sentence contains three verbs, and each verb has 25 corresponding semantic role tags. We do the same convolution operation on the tables of the same size.

The model architecture, shown in (Figure13) ,the input data of the model is semantic role relation tables(Figure 3~11), each figure undergoes the same convolution operation. The $k$ is the set of types of semantic role tags. Let $i \epsilon k$, the $i$-th represents a specific semantic role tag in a sentence. Let the v(padded where necessary) means the maximum number of verbs in a sentence. Let the s(padded where necessary) means the maximum number of sentences in the article.

A convolution operation involves a filter $w_1 \epsilon R^{vs}$, which is applied to a window of each semantic role relation table to produce a new feature.

$$c_i = f\left(w_1 \cdot x_{i:i+1} + b\right) \tag{1}$$

For example, a feature $c_i$ is generated from a window of a table. Here $b \epsilon R$ is a bias term, and f is a non-linear function such as the hyperbolic tangent. This filter is applied to each possible semantic role relation table window to produce a feature map.

$$\mathbf{c_1} = [c_1, c_2, \cdots, c_{25}] \tag{2}$$

Another convolution operation involves a filter $w_2 \epsilon R^{kvs}$, which is applied to a window of each semantic role relation table to produce a global feature.

$$\mathbf{c_2} = f\left(w_2 \cdot x_{1:25} + b\right) \tag{3}$$

$$g = \mathbf{c_1} \oplus \mathbf{c_2} \tag{4}$$

Where $\oplus$ is the concatenation operator to contact local features with global features.

$$\mathbf{g} = g_1 \oplus g_2 \cdots \oplus g_9 \tag{5}$$

These features extracted from each semantic role relation table are spliced. Then These features are passed to a fully connected softmax layer whose output is the probability distribution over labels.

# 3 EXPERIMENT

## 3.1 RESULT ANALYSIS

Table 1: Experiments with different maximum numbers of verbs in a sentence, different maximum numbers of sentences in a article.

| article_sent_num($\leq$) | verb_num($\leq$) | case_num | sent_num | Accuracy | EM | F1 |
|---|---|---|---|---|---|---|
| 5 | 3 | 1692 | 6848 | 0.7627 | 0.3965 | 0.7944 |
| 5 | 4 | 2279 | 9408 | 0.7600 | 0.3865 | 0.7825 |
| 6 | 3 | 2336 | 10752 | 0.7452 | 0.3116 | 0.7665 |
| 6 | 4 | 3262 | 15296 | 0.7533 | 0.3258 | 0.7617 |
| 7 | 3 | 2855 | 14400 | 0.7509 | 0.2959 | 0.7414 |
| 7 | 4 | 4075 | 20992 | 0.748 | 0.2743 | 0.7364 |
| 8 | 3 | 3225 | 17344 | 0.7557 | 0.275 | 0.7349 |
| 8 | 4 | 4673 | 25792 | 0.7527 | 0.2673 | 0.7271 |
| 9 | 3 | 3382 | 18816 | 0.7599 | 0.285 | 0.7172 |
| 9 | 4 | 5024 | 28928 | 0.7555 | 0.2555 | 0.7156 |
| 10 | 3 | 3521 | 20096 | 0.7615 | 0.274 | 0.7229 |
| 10 | 4 | 5219 | 30976 | 0.7559 | 0.2433 | 0.7132 |

We conduct experiments on the HotpotQA data set. HotpotQA encourages explainable QA models by providing supporting sentences for the answer, which usually come from several documents (a document is called "gold doc" if it contains supporting facts). Each case includes ten paragraphs, two of which are "gold doc." We only use "gold doc" as an article. Since the test set of HotpotQA is not publicly available, our evaluations are based on the dev set. We use exact match (EM), F1, Accuracy as three evaluation metrics.

The number of sentences and the number of verbs in sentences affects the semantic role feature map size. Therefore, we conducted experiments on the data of the maximum number of sentences (5~10) in an article and the maximum number of verbs (3~4) in a sentence on the proposed model, and the experimental results are shown in Table 1. The (article_sent_num) stands for the maximum number of sentences in an article. The (verb_num) stands for the maximum number of verbs in a sentence The (case_num) stands for the number of all the articles in the data sets that satisfy the constraint condition. The (sent_num) stands for the number of all the sentences in the data sets that satisfy the constraint condition.

From (table 1), as the number of sentences in articles increases, EM decreases, F1 decreases, and Accuracy does not change much. As the number of verbs in the sentence increases, EM, F1, and Accuracy do not change much.

There are some experiment result in table 2. From (table 2), within a certain range, EM does not

Table 2: Some experimental results in the number of sentences not more than 10, the number of verbs not more than 4. The case_num = 5219, the sent_num = 30976.

| article_sent_num($\leq$) | verb_num($\leq$) | Accuracy | EM | F1 |
|---|---|---|---|---|
| 10 | 4 | 0.7559 | 0.2433 | 0.7132 |
| 10 | 4 | 0.7621 | 0.2603 | 0.6939 |
| 10 | 4 | 0.7628 | 0.2638 | 0.6778 |

increase with the increase of F1. Sometimes, the EM decreases with the increase of F1.

Table 3: the result with Baseline and Bert.

| Model | case_num | Accuracy | EM | F1 |
|---|---|---|---|---|
| SAE-large | 7405 | - | 0.6330 | 0.8738 |
| Bert | 5219 | 0.8753 | 0.5214 | 0.8434 |
| Baseline | 7405 | - | 0.2195 | 0.6666 |
| Our model | 5219 | 0.7559 | 0.2433 | 0.7132 |

From (table 3), Although we did not use all the data in the dev set, we used most of the dev set. so experimental results can be compared to the Baseline. Our model performance should be close to the Baseline model. However, our model structure is simpler than Bert and Baseline. The Baseline

model reimplemented the architecture described in Clark & Gardner (2018) The Baseline model subsumes the technical advances on question answering, including word embedding, character-level models, bi-attention Seo et al. (2017) and self-attention Wang et al. (2017). Recently proposed models such as SAE Tu et al. (2020) include BERT module, so the performance is excellent. This shows that the semantic role relation table is a critical feature in machine reading comprehension.

## 3.2 LOW-RESOURCE EXPERIMENT

Table 4: The result in dev set. The case_num = 5219, the sent_num = 30976.

| article_sent_num($\leq$) | verb_num($\leq$) | train_case_num | Accuracy | EM | F1 |
|---|---|---|---|---|---|
| 10 | 4 | 250 | 0.6686 | 0.133 | 0.6682 |
| 10 | 4 | 500 | 0.7251 | 0.1941 | 0.6949 |
| 10 | 4 | 2500 | 0.7567 | 0.2381 | 0.6884 |
| 10 | 4 | 5000 | 0.7592 | 0.2507 | 0.6726 |
| 10 | 4 | 50000 | 0.7542 | 0.2476 | 0.7119 |

We do experiment on the training set and dev set(the number of sentences not more than 10, the number of verbs not more than 4). From (table 4), when the training set is lower than 2500 (5%), the performance(EM, Accuracy, and f1 ) is lower than that of the model using all training set. When the training set in 2500~5000 (5%~10%),the performance(EM, Accuracy) is close to the model using all training set. This shows that the semantic role relation table is a convenient feature, and the model's performance based on the semantic role relation table is stable.

## 3.3 ABLATION STUDIES

We do ablation experiments on the dev set(The number of sentences ($s$) not more than 10, the number of verbs ($v$) not more than 4. And The number of sentences ($s$) equal 5,the number of verbs ($v$) equal 3). Local(L): The model only uses the features, which are produced by the filter $w_1 \epsilon R^{vs}$.

Table 5: Ablation experiment

| model | case_num | Accuracy | EM | F1 |
|---|---|---|---|---|
| s=5,v=3(L+G) | 461 | 0.7382 | 0.2689 | 0.7275 |
| s=5,v=3(G) | 461 | 0.7335 | 0.2472 | 0.7037 |
| s=5,v=3(L) | 461 | 0.7339 | 0.2516 | 0.7201 |
| s$\leq$ 10,v$\leq$ 4(L+G) | 5219 | 0.7559 | 0.2433 | 0.7132 |
| s$\leq$ 10,v$\leq$ 4(G) | 5219 | 0.7592 | 0.2496 | 0.7012 |
| s$\leq$ 10,v$\leq$ 4(L) | 5219 | 0.7613 | 0.2525 | 0.6835 |

Global(G): The model only uses the features, which are produced by the filter $w_2 \epsilon R^{kvs}$.

From (table 5), when the training set is small, the performance(Accuracy, EM, F1) of the model(L) is significantly higher than the model(G). Moreover, the performance(Accuracy, EM, F1) of the model(L+G) is higher than the model(L) or the model(G). When the training set and the dev set increase, the performance(Accuracy, EM ) of the model(L+G) may be lower than than the model(L) or the model(G). This result shows that our model may not fully use local features, which are critical for EM.

## 3.4 CASE STUDIES

We do case studies experiments on the dev set(The number of sentences ($s$) not more than 5, the number of verbs ($v$) not more than 3. (G,L) is the same as (G,L) in ablation studies. A:all fine-grained features of semantic relational tables are input into the fully connected neural network. epoch:refers to the number of training. From (figure 14), the model(G+L) works best, and it is the model chosen for this paper. Model(G) and model(L) are similar in performance. Model(A) has poor performance,large fluctuation range. Because The model(A+G,A+L,A+G+L) contain too many fine-grained features, it is difficult to use effectively, and the performance lower than the model(L) and model(G).

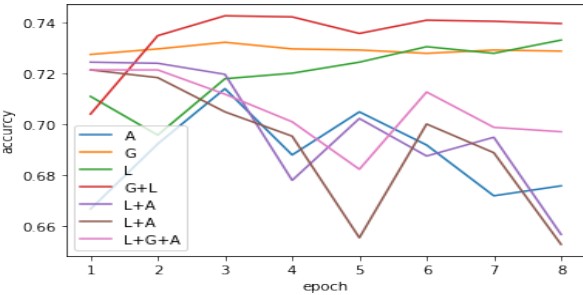

Figure 14: case studies

# 4 RELATED WORK

## 4.1 SUPPORTING FACT

Supporting factors is useful and becomes an important component in multiple-choice reading comprehension (Wang et al., 2019), natural language inference (Chen et al., 2017), open-domain question answering (Lin et al., 2018). Following HotpotQA , several benchmarks on open-domain tasks have gradually refined the supporting facts annotation, whose benefits have been demonstrated in terms of interpretability, bias, and performance (Dua et al., 2020; Inoue et al., 2020).

## 4.2 EXPLAINABLE MODEL

The study in Zhou et al. (2018) proposed an Interpretable Reasoning Network for QA on a knowledge base. The baseline model provided in the HotpotQA paper (Yang et al., 2018) and the QFE model Computational Linguistics (Nishida et al., 2019) are based on a single document MRC system proposed in Clark & Gardner (2018), with interpretable answer prediction. However, multi-hop reasoning was not explicitly dealt with in this work. Recent studies on multi-hop QA also build graphs based on entities and reasoning over the constructed graph using graph neural networks. DFGN Qiu et al. (2019) considered the model explainability by locating supporting entities and then leading to support sentences. The Hierarchical Graph Network (Fang et al., 2020) leverages a hierarchical graph representation of the background knowledge (question, paragraphs, sentences, and entities). SAE Tu et al. (2020) defines three edge types in the sentence graph based on the named entities and noun phrases appearing in the question and sentences. The Semantic table incorporates the semantic features of the entity graph (Battaglia et al., 2018)that uses sentence vectors as nodes and edges connecting sentences that share the same named entities and reflect the semantic role relation.

## CONCLUSION AND FUTURE WORK

In this paper, we propose the semantic role relation table form to enhance the interpretability of the processing process. We use the name of entities to integrate into the semantic role relation table. The model based on a semantic relational table achieves good performance in the experiments despite using a simple neural network structure. The model's performance based on the semantic role relation table is still stable with few training data sets. Unlike most recent works focusing on heuristically stacking complex mechanisms for performance improvement, this work is to shed some light on how even after fusing different kinds of semantic signals, the representation of semantic relations maintains good interpretability.

There are at least four potential future directions. First, The name of entities contains a few semantic features. However, the pre-trained embeddings have the advantage of capturing semantic similarity. Explore how to integrate the semantic signals of Pre-trained embeddings in a sematic role relation table to enhance the performance further. Second, according to our experimental results, to make full use of the information of the semantic relational table, it is necessary to propose a more effective neural network model. Third, SRL contains the shallow semantic information of all the words in a sentence. Therefore, it is convenient to integrate the finer-grained interpretable

semantic features(eg:the part-of-speech(POS)) of the text corresponding to SRL into the semantic relational table, to further enhance the interpretability of semantic relational tables. Forth, in some specific situations, the pronunciation of words and the image information related to words are also essential features.

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

## A  APPENDIX

You may include other additional sections here.

