# OpenReview forum: "Interpretable Semantic Role Relation Table for Supporting Facts Recognition of Reading Comprehension"
_ICLR.cc/2022/Conference — ICLR 2022 Submitted_

### Official Review · Reviewer_uT9c · 2021-10-28

**Correctness:** 3
**Technical Novelty And Significance:** 3
**Empirical Novelty And Significance:** 3
**Recommendation:** 5
**Confidence:** 3

**Details Of Ethics Concerns:**

I have no concerns.

**Main Review:**

Strengths

- Clear description of background knowledge, and clear exposition of the proposed model with examples.
- The authors perform an evaluation on  Question Answering and an ablation study.
- The findings show that the proposed model has comparable performance with related work.

Weaknesses
- It is not clearly discussed the evaluation of the model interpretability.
- It is not clear how the selection of hyper-parameters would affect the baseline compared to the addition of the semantic features.

Questions to the Authors. Please address the following questions during the rebuttal:

- Which evaluation can be used to highlight the model decisions given the input semantic features? Elaborate on the interpretability of the proposed features.
- Elaborate on the dependency of the proposed model on linguistic annotation. How does a SRL parser affect performance?
- Could you elaborate on the baseline set-up and the selection of the hyper-parameters?
- How sensitive the model is to a given set of hyper-parameters?
- How multiple  random runs could affect the final results?
- How the semantic features could be used on other downstream tasks? such as, Summarization or language inference.
-  The authors could observe more benefits with  using a graph neural network model than the current baseline. For example, De Cao et al. Question Answering by Reasoning Across Documents with Graph Convolutional Networks.

 Extra: please double check for typos and the title of the paper.


**Summary Of The Paper:**

The authors propose a method for adding interpretable semantic features for a Machine Reading Comprehension model  based on semantic relations across words and sentences.  The goal is to use semantic role labeling annotation to produce features used on a neural model for a language downstream task.  The main contributions are:  semantic features based on linguistic annotation, and interpretable semantic features for training a neural model. The study shows that the semantic features used to train Question Answering model performs comparable to related work.


**Summary Of The Review:**

The authors proposed the addition of linguistic annotation into a neural model for Question Answering. The paper clearly describes related work, and proposed model. The proposed model incorporates interpretability, however, it relies on manual or automatic semantic annotation.

---

### Official Review · Reviewer_aAdr · 2021-11-01

**Correctness:** 1
**Technical Novelty And Significance:** 2
**Empirical Novelty And Significance:** 1
**Recommendation:** 1
**Confidence:** 4

**Main Review:**

Strengths:

- The hypothesis that performing some kind of inference based on matching
  predicate argument structures of supporting sentences and question is intuitively appealing.

- The limited experiments presented seem to indicate that the method has
  some potential.


Weaknesses:

- The experiments are very preliminary, as they compare systems on different
  subsets of examples, use a baseline which is not clear whether it is
  comparable head-to-head to the proposed system, and the target system
  performs well below competing systems.

- The paper should be much more clear in the description of the motivating
  example and the method. In the current writing it cannot be understood
  in its entirety. There are many details missing and the explanations are
  very convoluted and vague.

- In order to show that the proposed technique captures information which is
  complementary to current state-of-the-art systems, the authors should try
  to incorporate their method in one of those systems.

In addition, the authors should study and review the latest related work,
clearly write the problem setting, model description, experimental setup and
results, as well as give an analysis of the results to provide an
understanding of why it is working.


**Summary Of The Paper:**

The paper proposes to use the predicate argument structure of sentences in
order to improve the results of a question answering system on the HotpotQA
dataset, where the answer depends on a number of supporting sentences.

The main contribution is a method encode whether the fillers of predicate arguments match
across supporting sentences and the question.



**Summary Of The Review:**


The lack of clarity and the weak empirical evidence are well below ICLR
standards, and thus I recommend a strong rejection.

---

### Official Review · Reviewer_dcWb · 2021-11-02

**Correctness:** 2
**Technical Novelty And Significance:** 2
**Empirical Novelty And Significance:** 2
**Recommendation:** 3
**Confidence:** 4

**Details Of Ethics Concerns:**

"Recently, Ribeiro et al. (2020) show that although measuring held-out accuracy has been the primary
approach to evaluate generalization, it often overestimates the performance of NLP models."
<= Here, "although measuring held-out accuracy has been the primary approach to evaluate generalization, it often overestimates the performance of NLP models." seems exactly the same as the first clause of the abstract in Ribeior et al., 2020




**Main Review:**

- Strengths: I think using semantic roles to enhance interpretability of a model is a good idea, since semantic roles provide important relation information.

- Weaknesses:
1) The writing needs to be improved a lot. Currently, it is hard to understand many parts and justifications are often missing.
2) The paper claims that using SRL can improve interpretability of a model. However, there is no supporting experiment showing that.

- Detailed feedback
1) I realized this work addresses the supporting sentence prediction task only after reading the beginning of 2.1. Before that, it was not clear to me, whether this work was proposing a module for interpretation that can be plugged in MRC models, proposing an MRC model that strengthens interpretability, or something else.
2) abstract: "better performance" than what?
3) Figure 1: "Paragh"? "Parapgh"? I guess they are typos of "paragraph".
4) Introduction: It is not clear to me how attention mechanisms are relevant to this work, even after reading the whole paper
5) Below Figure 2: references are missing for "the recent model", "We believe that the model ..."?
synonyms?
6) I see lots of typos. e.g., "and It is easy", wrong styles of citation,..
7) "the semantic signals"? What are they?
8) Why did you show the experiment with using smaller number of training examples? Does this say that the proposed model uses the semantic role relation even for improving the accuracy in small datasets?
9) "We arrange the different semantic roles labels in a fixed order" <= not sure what this means.
10) "set the position ... to 1" "set the position .. to 0" <= these descriptions are confusing. I think the authors set the value at a position in a table/matrix to 0/1.
11) "Who" does not appear in the figure due to it is regarded as a stop word? <= I think it should not appear in the table because it is what the question asks? I understand using stop words could be a heuristic way of getting it, though.
12)  the caption of Figure 13 didn't make sense to me.
13) "types"? What types" examples would be helpful.
14) typos: "let v means" "let s means" => "mean"
15) need to specify what x_i is.
16) global? local? how do you define them?
17) Why is the different maximum number of verbs in a sentence important? The paper says because it affects the feature map size. Why is it important?
18) How is "Exact Match" computed? How is it different from accuracy?
19) "target sentence" is not defined early enough.
20) what kind of classifier is used?
21) descriptions of baselines are too late.
22) "Although we did not use all the data in the dev set, we used most of the dev set. so
experimental results can be compared to the Baseline" <= this doesn't make sense to me. How can you compare the results from different algorithms when the data are different.
23) The last sentence before 3.2. What is this? The previous sentence talks about related work.
24) Why did you do the low-resource experiment?
25) Did you use data that only satisfy the condition of the maximum # of sentences and # of verbs? What is the average number of sentences and verbs in a sentence? Also, what happens if an article or a sentence includes more sentences or verbs?
26) I didn't understand what the case studies do and why.
27) How would synonyms be treated when matching the SRL tables among a question, target sentence, and articles?


**Summary Of The Paper:**

The paper presents a model for the supporting sentence prediction task in machine reading comprehension. To enhance interpretability of the model, the paper proposes to use the semantic relations among a question, target sentence, and sentences in an article as features of the model. The semantic relations are obtained from semantic role labeling and named entities. The paper shows the prediction accuracy of the proposed model compared against some prior models, and conducts ablation experiments and case studies.

**Summary Of The Review:**

I think the paper has too many unclear parts. In addition, the paper proposes using semantic role labeling to enhance interpretability, but I don't see how SRL enhances interpretability in the paper -- no experiments or studies about that.

---

### Official Review · Reviewer_GpZ5 · 2021-11-03

**Correctness:** 2
**Technical Novelty And Significance:** 2
**Empirical Novelty And Significance:** 2
**Recommendation:** 1
**Confidence:** 4

**Main Review:**


## Strengths

* The table-based featurization is interesting. It seems like a potentially nice way of characterizing propositional overlap, where different sentences have semantic substructures in common.

* It's nice that ablations were done and a study of how the model performs with less supervision

## Weaknesses

Unfortunately, I think this paper is a bit underdeveloped and there are several issues with the presentation.

* First, the model is not fully described. The paper gives a "simplified example" but never elaborates on it to describe how the full model works. I can't figure out, for example, how it is determined whether two arguments in different sentences should be considered to overlap — in the example they share the word "Tom", but how is it done? Just by token identity? If so, isn't this pretty problematic?

* Second, not enough background information is given about the task, dataset, and I/O spec of the model. Does this only apply to extractive QA? How are the metrics on HotpotQA calculated? Why are some of them missing? (Accuracy for SAE-large and Baseline in Table 3)

* Third, I think there are a lot of other issues which make the paper overall hard to understand.
  * Table 1 is an overload of numbers. Fewer significant figures are needed for the metrics and the trends in the table should be conveyed in a graph (or graphs).
  * snake_case variable names and other abbreviations are used throughout the paper where simple language would have sufficed and been easier to understand. It feels almost like directly describing the code — I think it's best to keep the reader isolated from unnecessary details. I have a hard time understanding Sections 3.3 and 3.4 for this reason. It's hard for me to remember what $w_2$ refers to, and I actually still don't intuitively understand it from its original description in Sec. 2.2.

* Fourth, I don't see what is interpretable or explainable about this model. While it is featurized in a way that depends on SRL, explainability of the model decisions would require a usable theory of how these features relate to the final decisions.

Finally, in light of all of these issues I don't know what to make of the results. They are apparently better than a different pre-BERT baseline but I don't understand the model description well enough to know what's better; and, it's not clear what insights we might take away from this work that could be applied to future models. Overall I think this paper presents an intriguing and promising approach, but needs a lot of organizational and presentation work (plus maybe more experimental work) before it is ready for scholarly publication.

## Minor comments

* I think I get what Figure 2 is getting at, but I don't think it effectively illustrates what an attention mechanism is.

* I do not understand how the conclusion at the end of Sec. 3.1 was drawn: it says "this shows that semantic role relation table is a critical feature in machine reading comprehension" just after describing how the baseline models perform well (Table 3).

* The low-resource study in Sec. 3.2 seems to me to show that performance levels off early, meaning semantic role relation tables are limited in what they can express alone.

* It seems to me that the binary featurization is basically just constructing a graph of nodes (for the tokens) and edges (for SRL relations and identity of roles and tokens). Isn't doing convolutions over this table something like just using a GCN? In that case, has this not been done before?

**Summary Of The Paper:**

This work proposes a model architecture for reading comprehension based on a large number of binary features corresponding to propositional overlap (in terms of SRL and, as far as I can tell, token surface forms or lemmas) between sentences in a passage and a reading comprehension question. These features are arranged into tables which are passed into a few convolutional filters to yield representations that are passed into a final classification layer.

**Summary Of The Review:**

Intriguing idea, but needs a lot of organizational work, clarity/presentation improvements, etc., and it's unclear what we learn from the results.

---

### Decision · Program_Chairs · 2022-01-20

**Decision:**

Reject

**Comment:**

All reviewers agree that this paper does not meet the bar for ICLR. The reviewers provide detailed feedback to the authors on how to improve the writing as well as the overall content of the paper.